# Region-aware Grasp Framework with Normalized Grasp Space for Efficient 6-DoF Grasping

**Siang Chen**[1*], **Pengwei Xie**[1*], **Wei Tang**[2], **Dingchang Hu**[1], **Yixiang Dai**[1], **Guijin Wang**[1,3,✉]

[1]Department of Electronic Engineering, Tsinghua University
[2]Shenzhen International Graduate School, Tsinghua University
[3]Shanghai AI Laboratory, China
[*]Equal Contribution

**Abstract:** A series of region-based methods succeed in extracting regional features and enhancing grasp detection quality. However, faced with a cluttered scene with potential collision, the definition of the grasp-relevant region stays inconsistent. In this paper, we propose Normalized Grasp Space (NGS) from a novel region-aware viewpoint, unifying the grasp representation within a normalized regional space and benefiting the generalizability of methods. Leveraging the NGS, we find that CNNs are underestimated for 3D feature extraction and 6-DoF grasp detection in clutter scenes and build a highly efficient Region-aware Normalized Grasp Network (RNGNet). Experiments on the public benchmark show that our method achieves significant $> 20\%$ performance gains while attaining a real-time inference speed of approximately 50 FPS. Real-world cluttered scene clearance experiments underscore the effectiveness of our method. Further, human-to-robot handover and dynamic object grasping experiments demonstrate the potential of our proposed method for closed-loop grasping in dynamic scenarios.

**Keywords:** 6-DoF Grasping, RGBD Perception, Normalized Space, Heatmap

## 1 Introduction

Grasping, a fundamental task for robotic manipulation, has seen significant advancements. Notably, recent strides in deep-learning methodologies have paved the way for data-driven grasp detection techniques. These techniques, capable of directly deducing the positions and rotations of available grasps, offer superior generalization to novel scenes compared to the traditional template-based methods. Meanwhile, 6-DoF grasping has grown significantly because of its effectiveness in applications and scenarios that require accurate and reasonable manipulation of objects.

Towards efficient 6-DoF grasping, most grasp detection methods treat the task as a detection-style problem involving the acquisition of optimal grasp positions and rotations from a single-view RGBD image or point cloud. Most of them process the global scene information to generate all possible grasps in the scene. Typical works like [1, 2] extract scene geometric features and regress grasps directly. The crucial problem is the redundancy and ambiguity of the entire scene information, which is time-consuming. Facing the challenge of grasping in cluttered scenes, recent methods [3, 4, 5, 6, 7, 8] encode global per-points features to infer potential grasp centers and segment graspable regions by integrating the regional points with the global features. While global-to-local schemes have demonstrated effectiveness, leveraging regional information without a consistent representation may undermine the generalization capability for novel scenes. The primary challenge lies in devising a method to extract regional information in a consistent representation.

In this work, we propose a novel **Region-aware Grasp Framework** decoupling grasp detection into graspable region extraction and follow-up regional grasp generation. Specifically, we propose **Nor-**

8th Conference on Robot Learning (CoRL 2024), Munich, Germany.

malized Grasp Space (NGS), a unified representation bridging the patch extraction and regional grasp detection. With extracted region information, the grasp detection problem within NGS can be formulated into a normalized form, which shows the invariance to the gripper size and the robustness to perturbations on the grasp center localization. Thus, considering grasp detection from a region-aware viewpoint, we propose **Regional Normalized Grasp Network (RNGNet)** to extract grasp-relevant geometric features and generate rotation heatmaps, demonstrating enhanced robustness and efficiency. Quantitative experiment results show a significant over 20% improvement in the grasp Average Precision (AP) on the public benchmark with $10\times$ faster inference speed than the former SOTA, AnyGrasp [9]. Real-world clutter clearance experiments validate our framework's robustness and generalizability. Further demonstration of grasping in dynamic scenes proves the efficiency and flexibility of our proposed region-aware framework and the potential for more complex tasks like target-oriented manipulation. The contributions of the paper mainly include:

1) A novel **Region-aware Grasp Framework** for 6-DoF grasp detection in cluttered scenes, decoupling grasp detection tasks into two-stage heatmap prediction and enhancing method efficiency.

2) **Normalized Grasp Space (NGS)**, a novel unified space defined for general parallel gripper grasp detection, empowering aligned and scale-invariant grasp detection in regions.

3) A highly efficient **Region Normalized Grasp Network (RNGNet)** aiming at extracting region-aware features and predicting high-quality 6-DoF grasps via rotation heatmap prediction.

## 2  Related Work

**Vision-based 6-DoF Grasp Detection:** Research has focused on detecting 6-DoF grasps primarily relying on single-view RGBD images or point clouds. Pioneer works ten Pas et al. [10], Liang et al. [11] mainly follow a sample-evaluation strategy, generating numerous grasp proposals and selecting optimal grasps through an evaluation model. Qin et al. [2] employ PointNet++ [12] for per-point feature extraction and direct regression of grasp parameters. Recent advancements, benefiting from large-scale datasets [3, 13, 14], adopt an end-to-end strategy. Wang et al. [6] introduces point-wise graspness to represent grasp location probabilities and aggregate regional features for rotation prediction. HGGD [15] employs 2D CNNs to generate heatmaps for grasp locations and integrate 2D semantics with 3D geometric features to configure 6D grasps. Tang et al. [16] investigates scene regions for high-quality grasp detection but ignores helpful regional characteristics. Diverging from existing approaches, our method employs a region-aware scheme forcing the network to capture scene-independent regional features, enhancing efficiency and generalization capability.

**Region Normalization for Grasp Detection:** Traditional 4-DoF grasp detection approaches [17, 18] conduct image-based normalization or crop the image patches with the exact sizes [19] to improve model performance. However, such 2D frameworks are limited, overlooking critical 3D geometric information and sensitive to camera viewpoints. For 6-DoF grasp detection, the utilization of regional 3D geometry is also not unique. However, most methods operate under a fixed gripper setting and extract local regions of predetermined sizes [2, 3, 6, 5]. These methods struggle to generalize or require model retraining when applied to grippers with different sizes or novel scenes. We normalize regions and corresponding grasps according to adjustable gripper sizes to obtain consistent 3D normalized grasp spaces, allowing more efficient grasp detection and adaptation for challenging applications, such as dynamic grasping.

## 3  Method

Similar to prior works [15, 20, 3], we focus on the problem of 6-DoF grasp detection of parallel grippers in clutter scenes from a single-view RGBD image $\chi \in \mathbb{R}^{H \times W \times 4}$ as input. Inspired by [15], we adopt the grasp representation as $\boldsymbol{g} = (x, y, z, \theta, \gamma, \beta, w)$, in which $(x, y, z)$ is the 3-DoF translations of grasp centers, $w$ is the grasp width and $(\theta, \gamma, \beta)$ compose the 3-DoF intrinsic rotation as Euler angles. $w$ is predicted within the range of $[0, w_{gripper}]$ to avoid collisions in clutter scenes. $(\theta, \gamma, \beta)$ are all constrained in $[-\frac{\pi}{2}, \frac{\pi}{2}]$.

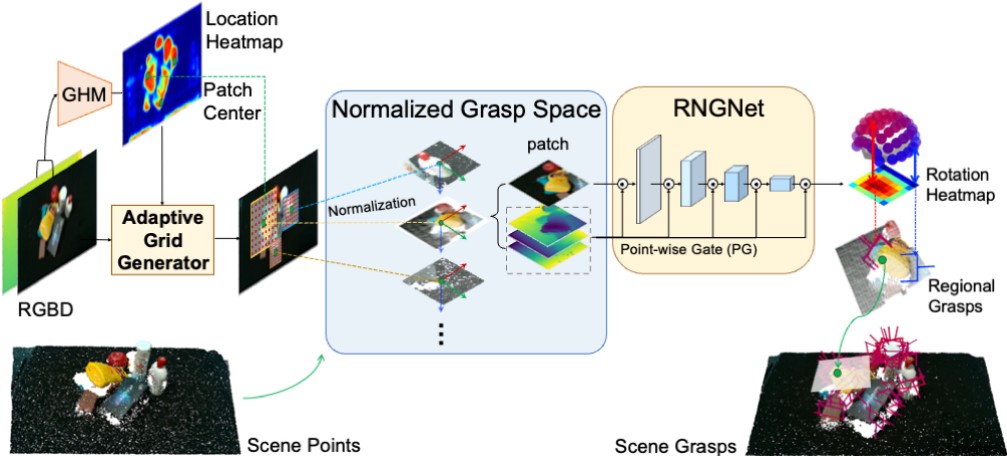

Figure 1: **Region-aware Grasp Framework.** With an RGBD image input, Grasp Heatmap Model (GHM) [15] predicts grasp location heatmap. Subsequently, graspable patches are extracted through an adaptive grid generator, which samples mesh grids based on the location heatmap and the gripper scale. Local patches are then converted into **Normalized Grasp Space** and fed into the **Regional Normalized Grasp Network** to predict regional rotation heatmaps and grasps. Finally, we apply the inverse of the normalization process and obtain the scene-level grasps.

## 3.1 Normalized Grasp Space

**Depth-Adaptive Patch Extraction:** As shown in Fig. 2(A)(B), we utilize the RGBD image $\chi \in \mathbb{R}^{H \times W \times 4}$ and patch centers $\mathbf{p}_i = (x_i, y_i, z_i)$ sampled from grasp location heatmap predicted by [15] as input, aiming at acquiring corresponding patches $\boldsymbol{P}_i^{raw} \in \mathbb{R}^{S \times S \times 4}$. To better leverage geometric features and avoid camera viewpoint variations, rather than using a fixed patch size, we design an adaptive grid generator that efficiently generates sampling grids with different sizes centered at the centers. Inspired by the setting in [21], the principle is to generate adaptive receptive fields for different patches and keep the receptive fields invariant to real-world distances. Thus, the patch sizes can be formulated as:

$$r_i = \frac{w_{ref} * F}{z_i},\qquad(1)$$

Figure 2: **Depth-Adaptive Patch Extraction and grasp representation.**

where $r_i$ is the patch image side length for the $i^{th}$ patch, $z_i$ is the depth to the $i^{th}$ center in the camera frame, and $F$ is the camera focal length. It is noteworthy that $w_{ref}$ is the reference receptive field size in the real world. To ensure the local patch size is adequate to infer grasps and avoid potential collisions, $w_{ref}$ is set to $2 * w_{gripper}$. Then the regular spatial grids will be rescaled and applied to the input RGBD image, resulting in the depth-adaptive local patches.

**Space Normalization:** Normalized Space has been proved helpful for neural network learning. During Space Normalization, our goal is to acquire a more structural and aligned representation among patches and regional grasps. Thus, firstly we convert the 4-channel raw RGBD image patches into the 6-channel RGBXYZ form $\boldsymbol{P}_i \in \mathbb{R}^{S \times S \times 6}$ using the camera intrinsics and imaging model, in which XYZ denotes the 3D coordinate maps. For clarity, because RGB channels require no extra normalization, we omit the RGB channels and only consider the XYZ channels in the following discussion. Then as is shown in Fig. 2(C), the normalized region patch $\boldsymbol{P}_i^*$ can be obtained with the patch centers $\mathbf{p}_i$ and the given reference receptive field $w_{ref}$ as below:

$$\boldsymbol{P}_i^* = \frac{\boldsymbol{P}_i - \mathbf{p}_i}{w_{ref}}.\qquad(2)$$

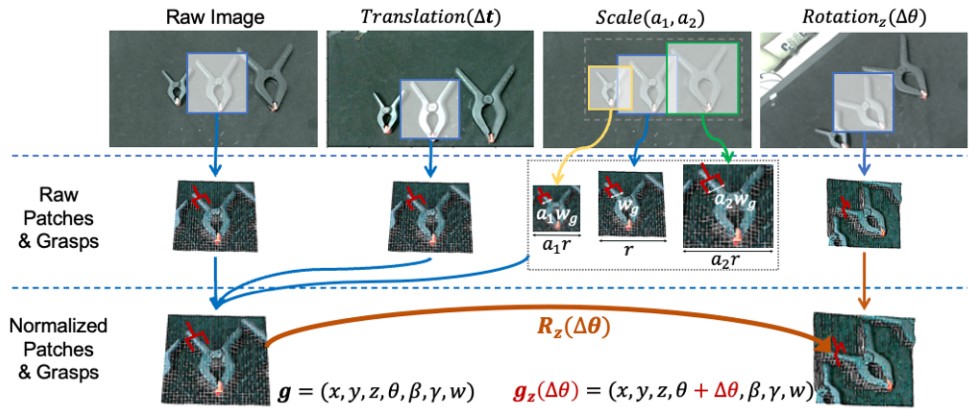

Figure 3: **Visualization for the characteristics of Normalized Grasp Space.**

Notably, input normalization is only part of the Normalization Grasp Space. We filter and normalize the grasp labels inside $\boldsymbol{P}_i^*$. Our 6-DoF grasp pose representation $\boldsymbol{g} = (x, y, z, \theta, \gamma, \beta, w)$ is equivalent to the representation in the SE(3) space $\boldsymbol{g} = (\mathbf{t}, \mathbf{R}, w)$, where $\mathbf{t} \in \mathbb{R}^{3 \times 1}$ denotes the translation, $\mathbf{R} \in \mathbb{R}^{3 \times 3}$ is the rotation matrix in the camera frame, $w$ denotes the grasp width. Thus, $\mathbf{R}$ can be represented as Euler angles: $\mathbf{R} = \mathbf{R}_z(\theta)\mathbf{R}_x(\beta)\mathbf{R}_y(\gamma)$, where $\mathbf{R}_z(\theta)$ represents the rotation of $\theta$ radians around the z-axis and similar notations apply for the other two rotations.

We limit our scope to grasp labels located within a spherical boundary of $0.1 * w_{ref}$ centered at $\mathbf{p}_i$. Such distance boundary is consistent with the widely adopted grasp coverage criterion in [22, 4, 15], in which two grasps are regarded as equivalent if their center distance is less than a threshold. The grasps, denoted as $\boldsymbol{G}_i^* = \{\boldsymbol{g}_j^*\}$, where $\boldsymbol{G}_i^*$ is the set of normalized grasp labels $\boldsymbol{g}_j^*$:

$$\boldsymbol{g}_j^* = (\mathbf{t}_j^*, \mathbf{R}_j^*, w_j^*) = (\frac{\mathbf{t}_j - \mathbf{p}_i}{w_{ref}}, \mathbf{R}_j, \frac{w_j}{w_{ref}}),$$
$$\boldsymbol{G}_i^* = \{\boldsymbol{g}_j^* | \left\|\mathbf{t}_j^*\right\| < 0.1\}. \tag{3}$$

As Fig. 2(D) indicates, our region-aware framework only focuses on the grasps near the patch centers, making the problem statement clearer and more concentrated. Then, for the regional 6-DoF grasp detection problem, the defined grasp function is $f : \boldsymbol{P}_i^* \to \boldsymbol{G}_i^*$. Our goal is learn a network $\hat{f}_\lambda : \boldsymbol{P}_i^* \to \hat{\boldsymbol{G}}_i^*$ with parameters $\lambda$ to fit $f$. By inverting the normalizing process, we can finally get the grasp predictions $\hat{\boldsymbol{G}}_i$ for this patch in the camera frame.

**Characteristics of Normalized Grasp Space:** As shown in Fig. 3, constraining the grasp detection problem in the Normalized Grasp Space brings beneficial characteristics as follows. For clarity, we omit the patch index $i$ and denotes the grasp labels within the transformed patches as $\boldsymbol{G}_\mathcal{T}^*$.

1) *Translation-invariance:* When the patch $\boldsymbol{P}_i$ translates with $\Delta\mathbf{t}$, we get the same normalized patch $\boldsymbol{P}_i^*$ and the grasps inside the patch keep invariant:

$$\boldsymbol{G}_\mathcal{T}^* = f(\boldsymbol{P}_\mathcal{T}^*) = f(\text{Norm}(\boldsymbol{P} + \Delta\mathbf{t})) = f(\text{Norm}(\boldsymbol{P})) = f(\boldsymbol{P}^*) = \boldsymbol{G}^*. \tag{4}$$

2) *SE(2)-rotation invariance* for $(\beta, \gamma)$ and *equivariance* for $\theta$*:* When the patch $\boldsymbol{P}_i$ rotates along the z-axis in the camera frame with $\Delta\theta$, the grasps inside the patch rotates equivariantly:

$$\mathbf{R}_\mathcal{T}^* = \mathbf{R}_z(\Delta\theta)\mathbf{R}^* = \mathbf{R}_z(\Delta\theta)\mathbf{R}_z(\theta)\mathbf{R}_x(\beta)\mathbf{R}_y(\gamma) = \mathbf{R}_z(\theta + \Delta\theta)\mathbf{R}_x(\beta)\mathbf{R}_y(\gamma). \tag{5}$$

3) *Scale-invariance:* With the coordinates (XYZ) of the patches or the scene scaled by a factor $a \in \mathbb{R}^+$, we get the same normalized patch $\boldsymbol{P}_i^*$ and the grasps inside the patch keep invariant if the receptive field for normalization is scaled by the same factor $a$:

$$\boldsymbol{G}_\mathcal{T}^* = f(\boldsymbol{P}_\mathcal{T}^*) = f(\text{Norm}(a\boldsymbol{P}|w_{ref} = aw_0)) = f(\frac{a(\boldsymbol{P} - \mathbf{p}_i)}{aw_0}) = f(\boldsymbol{P}^*) = \boldsymbol{G}^*. \tag{6}$$

Thus, based on the characteristics above, the latter trained grasp function $\hat{f}_\lambda$ to fit the ground truth grasp function $f$ is able to generate consistent results across scenes and grippers in different scales.

**Patch Scale Randomization:** Above, we consider the ideal scenario for applying Normalized Grasp Space. However, in the real world, noise from depth sensors is non-negligible and may cause unstable patch extraction. Furthermore, the gripper sizes in grasp detection datasets are usually fixed, which means that though we train our model through a normalized scheme, it senses the regions with a fixed receptive field. Thus, we add randomization to the receptive field $w_{ref}$ during training. The receptive field is randomly selected in the range $[\frac{w_{ref}}{2}, 2 * w_{ref}]$, which can improve the robustness and generalization capability of our grasp network.

### 3.2 Regional Normalized Grasp Network

**Region-aware Patch Feature Extraction:** To efficiently detect grasps in regions, our approach extracts patch-wise features in a region-aware manner. The XYZ coordinates of the patches are crucial for understanding object contact and collision. Hence, it is imperative to incorporate the overall 3D XYZ coordinates instead of only the depth channel for feature extraction. For geometric feature extraction from point-wise coordinates, previous methods utilize shared MLP (Multi-layer Perceptron) combined with sampling and grouping [23, 12, 24], which is effective but introduces a significant computation overhead. Thus, combining the advantages of point-wise shared MLPs and CNNs, as depicted in Fig. 4, we design an efficient network to process the normalized patches.

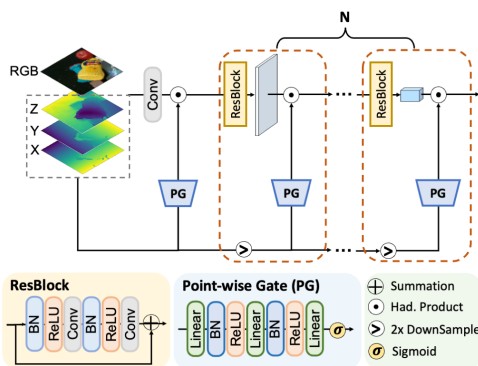

Figure 4: **Detailed structure of RNGNet.**

Inspired by the gated CNN in [25, 26], we establish a multi-stage Gated CNN with a novel Point-wise Gate (PG) Module. Point-wise Gate Module is built with shared MLP operating on 3D point coordinate XYZ maps, increasing the dimensions of XYZ maps in different resolutions to obtain spatial feature gate values in each convolution stage. By multiplying the original feature maps with the gate values, Point-wise Gate enables selective amplification or attenuation of filters within convolution layers.

**Rotation Heatmap Predictor:** Based on the regional anchor-based grasp generator in [3, 6, 15], with the region-aware features extracted, we predefine a series of non-uniform rotation anchors for $(\gamma, \beta)$ generated via the Anchor Shifting algorithm in [15], whose Cartesian product forms a 2D rotation heatmap (one axis for $\gamma$, the other of $\beta$). Then, the grasp detection can be considered a two-class semantic segmentation problem, while the normalized grasp widths and extra center offsets are regressed for each possible rotation (pixel in the heatmap).

## 4 Dataset Experiments

**Dataset and Metrics:** GraspNet-1Billion [3] is a large-scale grasp dataset widely adopted in recent 6-DoF grasp detection research [9, 15, 8, 20], which provides large-scale training data and a standard evaluation platform for the task of general robotic grasping. During our experiments, we follow the official evaluation pipeline and code of the GraspNet-1Billion dataset, in which detected grasp poses are first filtered with non-maximum suppression, and then the top 50 grasp poses are evaluated with force-closure [28] metrics under a series of friction coefficients condition.

**Quantitative Results:** We compare the overall performance of our RNGNet with multiple typical grasp detection algorithms trained on the same dataset training split. As Table 1 indicates, RNGNet achieves excellent performance, reaching an average 58.06/52.24 AP by RealSense/Kinect cameras.

| Method | Average ↑ | Seen ↑ | Similar ↑ | Novel ↑ | Paras ↓ | Time[1]/ms ↓ |
|---|---|---|---|---|---|---|
| GPD [10] | 17.48 / 19.05 | 22.87 / 24.38 | 21.33 / 23.18 | 8.24 / 9.58 | - | - |
| PointnetGPD [11] | 19.29 / 20.88 | 25.96 / 27.59 | 22.68 / 24.38 | 9.23 / 10.66 | - | - |
| GraspNet-baseline [3] | 21.41 / 23.08 | 27.56 / 29.88 | 26.11 / 27.84 | 10.55 / 11.51 | - | 126[†] |
| TransGrasp [27] | 27.65 / 25.70 | 39.81 / 35.97 | 29.32 / 29.71 | 13.83 / 11.41 | - | - |
| HGGD[†] [15] | 42.79 / 39.79 | 58.35 / 56.85 | 47.93 / 43.93 | 22.10 / 18.59 | 3.42M[†] | 34[†] |
| Scale Balanced Grasp [8] | 44.85 /  - | 58.95 /  - | 52.97 /  - | 22.63 /  - | - | 242[†] |
| GSNet [6] | 47.92 / 42.53 | 65.70 / 61.19 | 53.75 / 47.39 | 24.31 / 19.01 | 15.4M[†] | 61[†] |
| AnyGrasp *w/* CD [9] | 49.01 /  - | 66.12 /  - | 56.09 /  - | 24.81 /  - | 24.7M[†] | 198[†] |
| RNGNet | 58.06 / 52.24 | 75.20 / 72.23 | 66.62 / 58.43 | 32.38 / 26.05 | 3.66M | **17** |
| RNGNet *w/* CD | **59.13 / 52.81** | **76.28 / 72.89** | **68.26 / 59.42** | **32.84 / 26.12** | 3.66M | 20 |

"-": Result Unavailable; **CD:Collision Detection**, the grasp post-processing algorithm proposed in [3] and utilized by [9].
[1] Evaluated with $batch\_size = 1$ on Ubuntu20.04 with AMD 5600x CPU and a single NVIDIA RTX 3060Ti GPU.
[†] Reimplemented or tested with the provided codebases.     More detailed results are shown in Table 7 of the Appendices.

Table 1: **Results on GraspNet dataset.** Showing APs on RealSense/Kinect split and model efficiency.

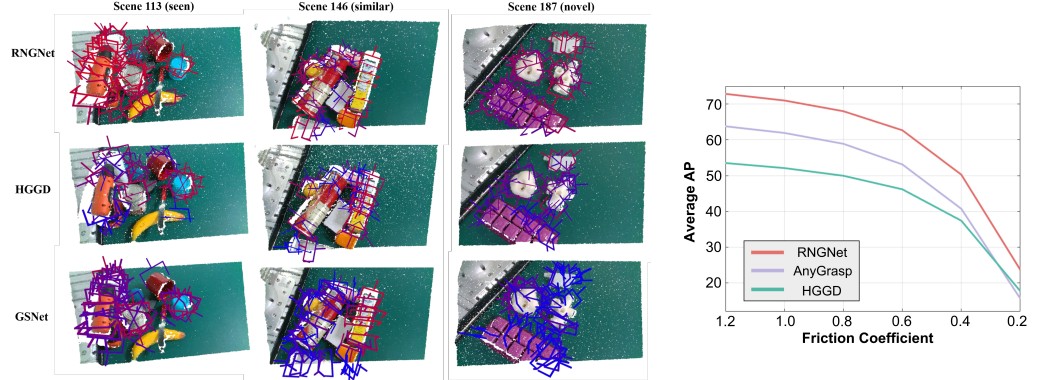

Figure 5: **Qualitative results covering seen/similar/novel test set.** Top 50 grasps after grasp-NMS [3] are displayed. Color implies the predicted grasp confidence (red: high, blue: low).

Figure 6: **Performance (Average AP) curve.** Tested under different difficulties (friction coefficient).

Compared with other methods, RNGNet outperforms current state-of-the-art by a large margin on all dataset splits, especially on the unseen (similar and novel) splits, proving the efficacy and generalizability of our proposed region-aware grasp framework. Furthermore, considering the computation cost for one shot of grasp detection, our method achieves the fastest speed with the best grasp detection quality, which implies the potential of our approach to be applied in dynamic scenes or on widely used computation-resource-constrained platforms in robot deployment.

As shown in Fig. 6, we conduct grasp evaluation using the force-closure metric under different difficulties with different object friction coefficients. With lower object friction, the object is more likely to drop from the gripper and thus requires more high-quality grasp detection results. It can be clearly seen in the figure that RNGNet surpasses former SOTAs by a large margin with different object friction coefficients, especially for the settings with lower friction coefficients.

As is indicated in Table 2, following the setting in [8], we evaluate the grasp quality at all scales (S: Small, M: Medium, L: Large). RNGNet significantly improves performance with all the metrics, especially in the small-scale grasps and the novel split, proving the proposed NGS's effectiveness. RNGNet can learn more robust and generalizable region features across different scales with NGS.

| Method | Seen | | | Similar | | | Novel | | |
|---|---|---|---|---|---|---|---|---|---|
| | $AP_S$ | $AP_M$ | $AP_L$ | $AP_S$ | $AP_M$ | $AP_L$ | $AP_S$ | $AP_M$ | $AP_L$ |
| Scale Balanced Grasp [8] | 13.47 | 48.12 | 61.81 | 6.23 | 37.90 | 53.89 | 7.60 | 17.04 | 23.10 |
| Scale Balanced Grasp + OBS [8] | 18.29 | 52.60 | 64.34 | 10.03 | 42.77 | 57.09 | 9.29 | 18.74 | 24.36 |
| RNGNet | **22.94** | **64.06** | **68.50** | **19.34** | **59.55** | **64.58** | **21.51** | **29.84** | **28.57** |

Table 2: **Multi-scale results.** Grasps with different scales are tested separately on RealSense split. OBS: Object Balanced Sampling proposed in [8] which requires extra instance segmentation.

| Components | | | Seen | Similar | Novel |
|---|---|---|---|---|---|
| *Norm* | *Adapt* | *Rand* | | | |
| | | | 66.71 | 51.66 | 26.97 |
| ✓ | | | 69.93 | 53.15 | 28.71 |
| ✓ | ✓ | | 73.98 | 64.89 | 30.99 |
| ✓ | ✓ | ✓ | **75.20** | **66.62** | **32.38** |

*Norm*: Patch Normalization
*Adapt*: Depth-adaptive Patch Extraction
*Rand*: Scale Randomization

| Components | | | | Seen | Similar | Novel |
|---|---|---|---|---|---|---|
| Z | XY | RGB | *PG* | | | |
| ✓ | | | | 72.68 | 61.49 | 28.31 |
| ✓ | ✓ | | | 73.76 | 63.43 | 30.15 |
| ✓ | ✓ | ✓ | | 74.31 | 65.41 | 31.72 |
| ✓ | ✓ | ✓ | ✓ | **75.20** | **66.62** | **32.38** |

*PG*: Point-wise Gate Module

Table 3: **Normalized Grasp Space ablation.** Tested on Realsense Split.

Table 4: **Input modality and network ablation.** Tested on Realsense Split.

**Qualitative Results:** We also conduct grasp detection visualization with HGGD [15] and GSNet [6] on three parts of the test set (seen: scene100-129, similar: scene130-159, novel: scene160-189). As depicted in Fig. 5, RNGNet exhibits superior grasp detection quality and achieves a higher grasp coverage rate in cluttered scenes than prior works, even in scenarios involving occlusion or partial observation. This enhancement provides the robot with more feasible action alternatives, benefiting scene-level grasp planning and execution. Furthermore, when compared with [15, 9], it is noteworthy that our proposed region-aware framework enables the generation of grasps better aligned with object centers and surfaces, consequently enhancing grasp stability. The overall qualitative results underscore the validity of our representation and approach in cluttered scenes, offering a more comprehensive perspective.

**Normalized Grasp Space Ablation:** We perform ablation studies on our proposed Normalized Grasp Space, which primarily consists of Patch Normalization, Depth-adaptive Patch Extraction, and Scale Randomization. The results in Table 3 illustrate the effects of these three components on Normalized Grasp Space. As expected, both Patch Normalization and Depth-adaptive Patch Extraction significantly enhance performance across all scenes, proving the effectiveness of the novel viewpoint about generating grasps in the region-aware and grasp-centric spaces. Additionally, Scale Randomization also boosts the performance of our methods on the dataset.

**Region Normalized Grasp Network Ablation:** Table 4 presents the ablation experiments conducted on the RNGNet with different input modalities. Remarkably, our method achieves a commendable performance even with only depth maps as input. Introducing XY maps or RGB images as supplementary inputs leads to considerable improvements. While geometric information from Z maps plays a crucial role in 6-DoF grasp detection, the additional positional information from XY maps and the inclusion of colors contribute to enhanced feature extraction and are especially beneficial for generalization to previously unseen (similar and novel) scenes. Finally, applying Point-wise Gate Module to feature maps in all stages yields further improvements across all test splits, with a negligible increase in the number of parameters (approximately 36k, or about 1% of the total parameters). These findings support the efficacy of our approach, which is built with a combination of point-based networks and CNNs.

## 5   Realworld Experiments

**Cluttered Scene Clearance:** To conduct a comprehensive comparison of grasp detection methods in real-world cluttered scenes, we constructed multiple cluttered scenes consisting of 6 to 9 randomly placed objects within a $50cm \times 50cm$ workspace, including novel objects, object stacking and occlusion, as shown in Fig. 7. We deployed multiple representative grasp detection methods [15, 9] on our real-world robot platform. Subsequently, akin to the metrics utilized in [15], we assessed the overall **Success Rate** and **Completion Rate** to facilitate overall comparison.

Compared with other methods, results presented in Table 5 demonstrate the generalization capability to real-world grasping of our framework. However, it is worth noting that HGGD struggles to generate grasps for objects with small scales, and AnyGrasp exhibits imprecise rotation predictions, resulting in unstable grasping and frequent drops, particularly evident when objects are stacked. Both methods are prone to failure when the captured point cloud becomes more noisy or only partial

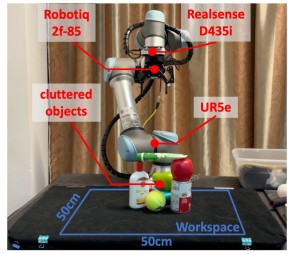

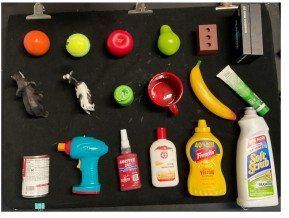

Figure 7: **Robot settings and object samples.**

| Scene | Objects | RNGNet | HGGD | AnyGrasp |
|---|---|---|---|---|
| 1 | 8 | 8 / 9 | 7[†] / 9 | 8 / 10 |
| 2 | 7 | 7 / 9 | 7 / 9 | 7 / 9 |
| 3 | 9 | 9 / 10 | 9 / 11 | 9 / 10 |
| 4 | 6 | 6 / 6 | 6 / 6 | 6 / 6 |
| 5 | 7 | 7 / 8 | 7 / 9 | 7 / 9 |
| 6 | 8 | 8 / 9 | 8 / 10 | 8 / 10 |
| 7 | 9 | 9 / 9 | 9 / 11 | 9 / 11 |
| **Success Rate** | | **90%**(54 / 60) | 82%(53 / 64) | 83%(54 / 65) |
| **Completion Rate** | | **100%**(7 / 7) | 86%(6 / 7) | **100%**(7 / 7) |

Table 5: **Real-world clutter clearance results.** Robot performs grasp detection and executes grasping for each scene until no grasp is detected or 15 attempts are tried. [†] means failed to grasp all the objects.

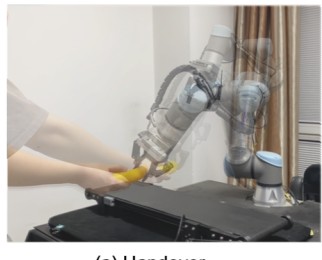

(a) Handover

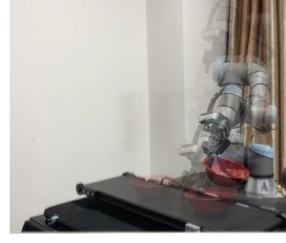

(b) Conveyor

Figure 8: **Real-world closed-loop experiment settings.**

| Setting | Success Rate |
|---|---|
| Handover | **82.5%** (33 / 40) |
| Conveyor | **67.5%** (27 / 40) |

Table 6: **Closed-loop results.**

object observations are available. In contrast, RNGNet, with its robustness to noise originating from the depth sensor, demonstrates the ability to predict more stable grasps with precise rotations. This effectively reduces collisions and object drops, showcasing the potential of this method.

**Closed-loop Grasping:** With RNGNet, we are able to detect grasps in any region regardless of the patch location or the scene structure. This implies a practical downstream application potential when there is no need to generate grasps across the scene. To evaluate the efficiency of our framework in more challenging scenes, we build two dynamic settings as illustrated in Fig. 8: (a) Human-to-robot handover; (b) Moving object on the conveyor. Experiment results in Table 6 confirm that our framework can accomplish dynamic grasping at a satisfactory success rate by detecting high-quality grasps in a real-time frame rate and executing the closed-loop controlling. The proposed closed-loop grasping pipeline can effectively handle more complex nontable-top grasping scenarios. Thus, our proposed method proves robust to a much noisier environment, which underscores the potential of our method for addressing challenging dynamic scenarios. The closed-loop grasping algorithm is demonstrated in Algorithm. 1 of the Appendices, and more demonstrations of the dynamic grasping experiments can be found in the Supplementary Videos.

## 6 Limitation and Conclusion

Normalized Grasp Space (NGS), a unified space for region-based grasp detection, is proposed to obtain consistent region representation. With the representation, we further build an efficient convolution-based Region Normalized Grasp Network (RNGNet) to extract features and predict rotation heatmap. Though efficient, our method, relying on single-view images, is vulnerable to noises and can be only applied with rigid bodies. We plan to explore integrating scene-level information from the multi-view image during the process of closed-loop grasping, which is critical for precise manipulation in complex scenes or with specific instructions. More limitations are discussed in the Section C and D of the Appendices.

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

# Appendices

## A  Additional Results

**Detailed Results on the GraspNet-1Billion Dataset:** In addition to the overall grasp AP (Average Precision) metric, we also provide the evaluated grasp quality based on force-closure metric [28] with varying friction coefficients, denoting $AP_\mu$, representing the average precision at given friction coefficient $\mu$. As is shown in Table 7, our proposed RNGNet outperforms previous state-of-the-art methods in terms of all metrics under different friction conditions, especially on the difficult scenarios (similar/novel splits and $AP_{0.4}$ with lower friction coefficient).

| Method | Seen | | | Similar | | | Novel | | |
|---|---|---|---|---|---|---|---|---|---|
| | $AP$ | $AP_{0.8}$ | $AP_{0.4}$ | $AP$ | $AP_{0.8}$ | $AP_{0.4}$ | $AP$ | $AP_{0.8}$ | $AP_{0.4}$ |
| GPD [10] | 22.87/24.38 | 28.53/30.16 | 12.84/13.46 | 21.33/23.18 | 27.83/28.64 | 9.64/11.32 | 8.24/9.58 | 8.89/10.14 | 2.67/3.16 |
| PointnetGPD [11] | 25.96/27.59 | 33.01/34.21 | 15.37/17.83 | 22.68/24.38 | 29.15/30.84 | 10.76/12.83 | 9.23/10.66 | 9.89/11.24 | 2.74/3.21 |
| GraspNet-baseline [3] | 27.56/29.88 | 33.43/36.19 | 16.95/19.31 | 26.11/27.84 | 34.18/33.19 | 14.23/16.62 | 10.55/11.51 | 11.25/12.92 | 3.98/3.56 |
| TransGrasp [27] | 39.81/35.97 | 47.54/41.69 | 36.42/31.86 | 29.32/29.71 | 34.80/35.67 | 25.19/24.19 | 13.83/11.41 | 17.11/14.42 | 7.67/5.84 |
| HGGD[†] [15] | 58.35/56.85 | 66.54/64.60 | 55.96/52.94 | 47.93/43.93 | 56.91/52.73 | 41.86/36.88 | 22.10/18.59 | 27.37/22.98 | 14.31/11.91 |
| Scale Balanced Grasp [8] | 58.95/ - | 68.18/ - | 54.88/ - | 52.97/ - | 63.24/ - | 46.99/ - | 22.63/ - | 28.53/ - | 12.00/ - |
| GSNet [6] | 65.70/61.19 | 76.25/71.46 | 61.08/56.04 | 53.75/47.39 | 65.04/56.78 | 45.97/40.43 | 23.98/19.01 | 29.93/23.73 | 14.05/10.60 |
| AnyGrasp w/ CD [9] | 66.12/ - | 77.27/ - | 61.02/ - | 56.09/ - | 68.29/ - | 47.31/ - | 24.81/ - | 31.08/ - | 13.82/ - |
| RNGNet | 75.20/72.23 | 85.02/82.04 | 71.71/67.95 | 66.62/58.43 | 78.53/68.69 | 59.23/51.87 | 32.38/26.05 | 40.44/32.29 | **19.84/16.24** |
| RNGNet w/ CD | **76.28/72.89** | **86.58/83.10** | **72.26/68.11** | **68.26/59.42** | **80.86/70.16** | **60.01/52.13** | **32.84/26.12** | **41.07/32.45** | 19.68/15.92 |

"-": Result Unavailable, CD: Collision Detection
[†] Reimplemented with official codebase without CD.

Table 7: **Detailed results on GraspNet Dataset.** Showing APs on RealSense/Kinect split under different friction coefficient.

**Extra Real-world Results:** For more detailed performance evaluation, we conduct extra real-world experiments at different noise levels to illustrate the practical benefits and robustness of our method. In the extra real-world experiments, for more challenging grasping, we introduce more novel objects (50 objects in total vs 24 used in the manuscript) in the experiments, and more objects (6 to 12 now vs 6 to 9 in the manuscript) may be randomly placed in a clutter. Each method is tested with extra 6 scenes under different settings. The other settings (grasp order, collision check, robot control, etc.) are kept the same as in HGGD [15] and the manuscript.

| Noise Deviation (m) | Method | Attempt Success Rate | Scene Clearance Rate |
|---|---|---|---|
| 0 | AnyGrasp | 52 / 67 = 77.6 % | 5 / 6 = 83.3 % |
| | RNGNet | 53 / 60 = **88.3** % | 6 / 6 = **100** % |
| 0.01 | AnyGrasp | 41 / 89 = 46.1 % | 2 / 6 = 33.3 % |
| | RNGNet | 48 / 80 = **60.0** % | 4 / 6 = **66.7** % |
| 0.02 | AnyGrasp | 8 / 90 = 8.9 % | 0 / 6 = 0 % |
| | RNGNet | 39 / 88 = **44.3** % | 2 / 6 = **33.3** % |

Table 8: **Additional Real-world Clutter Clearance Experiments.**

As is shown in the table above, our method achieves an impressive success rate in attempts without additional noise, efficiently clearing all scenes. In comparison, AnyGrasp exhibits lower attempt success and scene clearance rates and may struggle to clear scenes due to repeated failures on specific objects. Our method, on the other hand, offers more viable grasps across scenes with multiple sampled patches, which benefit the scene clearance process. This is also illustrated in Figure 5 of the manuscript (AnyGrasp mainly uses GSNet for one-frame grasp detection).

As for grasp detection with noisy inputs, AnyGrasp shows a significant decline in performance as noise increases, while our method performs more consistent grasp detection results in noisy environments and is capable of clearing scenes with relatively high noise levels (0.02 m). These results

demonstrate the superior robustness and reliability of our method across varying noise levels. Its ability to maintain high performance in the presence of noise makes it a more reliable choice, especially in environments where noise is a factor.

**Real-world Ablations:** One of the purposes of the proposed Normalized Grasp Space (NGS) is to better leverage geometric features and avoid camera viewpoint variations, keeping the receptive fields invariant to real-world camera positions. Thus, we conduct more extensive testing across various camera viewpoint to ablate the proposed NGS in the manuscript. We conduct robot grasping with a variable camera viewpoint rather than a fixed camera position by changing the camera distance to the table. All other experiment settings are the same as above.

| Camera Viewpoint | Method | Attempt Success Rate | Scene Clearance Rate |
|---|---|---|---|
| close: 0.5 m (default) | RNGNet *w/o* NGS | 52 / 66 = 78.8 % | 5 / 6 = 83.3 % |
| | RNGNet | 53 / 60 = **88.3** % | 6 / 6 = **100** % |
| medium: 0.6 to 0.8 m | RNGNet *w/o* NGS | 48 / 79 = 60.8 % | 4 / 6 = 66.7 % |
| | RNGNet | 53 / 62 = **85.5** % | 6 / 6 = **100** % |
| far: 0.8 to 1.0 m | RNGNet *w/o* NGS | 18 / 90 = 20.0 % | 0 / 6 = 0 % |
| | RNGNet | 53 / 65 = **81.5** % | 6 / 6 = **100** % |

Camera Viewpoint: camera distance to the table surface
Attempt Success Rate = Successful Attempt Number / Total Attempt Number
Scene Clearance Rate = Cleared Scene Number / Total Scene Number

Table 9: **Real-world Clutter Clearance Ablations of proposed Normalized Grasp Space.**

As the results show, RNGNet with NGS demonstrates superior robustness and adaptability across different camera viewpoints. The inclusion of NGS significantly enhances the method's ability to maintain high performance, even as the camera viewpoint moves further away. Without NGS, the performance drops considerably, especially at greater distances, making the NGS component crucial for reliable and consistent performance in varying conditions.

**Backbone Ablation:** To further prove the efficiency and robustness of our 2D-convolution-based network, in Table 10, we replace the backbone with the widely used [6, 20, 29] high-dimensional sparse convolution backbone, MinkowskiFCNN [30]. As is shown in the results, our simple 2D convolution network is more efficient than the more complicated sparse convolution network in experiments, especially with high-resolution input. When facing low-resolution input, SparseCNN performs slightly better due to its particular design for sparse input. Significantly, both RNGNet and SparseCNN show quite consistent performance with input patches of different resolutions, which proves the robustness of our region-aware framework.

| Backbone | Size | Seen | Similar | Novel | Time/ms |
|---|---|---|---|---|---|
| SparseCNN [30] | $64 \times 64$ | 72.17 | 61.90 | 29.85 | 74 |
| | $32 \times 32$ | 71.76 | 61.63 | **29.89** | 29 |
| | $16 \times 16$ | **68.16** | **56.68** | **25.76** | 18 |
| RNGNet | $64 \times 64$ | **75.20** | **66.62** | **32.38** | **8** |
| | $32 \times 32$ | **72.13** | **62.81** | 29.86 | **5** |
| | $16 \times 16$ | 67.04 | 55.35 | 25.63 | **4** |

Table 10: **Backbone ablation.** Showing APs on Realsense split and network inference time.

**Guidance Ablation:** Though RNGNet utilizes the Grasp Heatmap Model as a part of the patch extractor by default. However, we only introduce the Grasp Heatmap Model for locating graspable regions, and our model can be easily used without the Grasp Heatmap Model. To better investigate the ability of our method to detect high-quality grasp in regions, we compare the performance of our method under different settings (with or without Grasp Heatmap Model). Without the Grasp

Heatmap Model, we filter the table pointcloud by z coordinates and use farthest sampling on the foreground pointcloud to acquire region centers. As is shown in the table below, even without the guidance of the Grasp Heatmap Model, our method can still perform high-quality grasp detection for a relatively slower speed (more patches are processed).

| Localization | Seen | Similar | Novel | Time[1] (ms) |
|---|---|---|---|---|
| Heatmap | **75.20** | **66.62** | 32.38 | **17** |
| Foreground | 74.75 | 66.06 | **33.59** | 30 |

[1] Same settings as in Table 1.

Table 11: **Ablation Experiments of Patch Localization Algorithm.** Showing APs on Realsense split.

**Grasp Inference with Multi-scale Grippers:** As clarified in Section. 3.1, the Normalized Grasp Space (NGS) demonstrates scale invariance. In other words, our grasp detection model can be easily scaled to accommodate a different range from the fixed gripper width defined in the training dataset without any extra training. In Fig. 9, we illustrate this by enlarging the gripper scale from $w_{gripper} = 0.05\ m$ to $w_{gripper} = 0.2\ m$, showcasing grasps in a cluttered scene. These results affirm that our framework can seamlessly adapt to grippers of varying sizes. As is shown in Fig. 10, we extend the application of our framework to a much larger indoor scene than the typical table-top settings. The proposed RNGNet successfully generates large-scale grasps for large objects, validating the effectiveness of the Normalized Grasp Space.

To further investigate the effectiveness of our Normalized Grasping Space for object grasping with grippers of different scales, We build scenes with objects with significantly different scales in the Sapien [31] simulator and conduct grasp detection and robot grasping. Experiments prove that our Normalized Grasping Space can generalize well to grippers with different scales and has the potential to apply to industrial scenes, in which data and labels are hard to obtain. Detailed results and visualization can be found in our Supplementary Video.

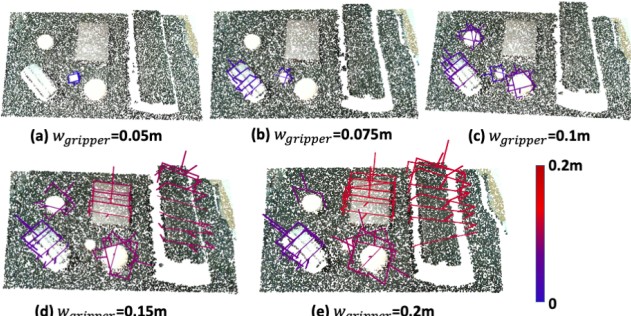

Figure 9: **Visualizations of grasp predictions of different gripper scales.** Different colors mean different gripper widths. From (a) to (e), as the gripper scale enlarges, it shows adaptive grasp predictions under different configurations.

## B    Implementation Details

**Normalized Patch Dataset Preparation:** Currently, most grasp datasets focus on providing object-level grasp labels [13, 14] or scene-level grasp label [3, 32], which are not directly suitable for our normalized scheme training. Although GraspNet-1Billion offers a large number of RGBD images and grasps annotations, based on the proposed Normalized Grasp Space, we need to construct a patch-based grasp dataset providing normalized patches and grasps annotations. As is shown in Fig. 11, utilizing the scene-level annotations provided in the GraspNet-1Billion dataset, we generate a new patch-based grasp dataset for Regional Normalized Grasp Network training. Firstly we

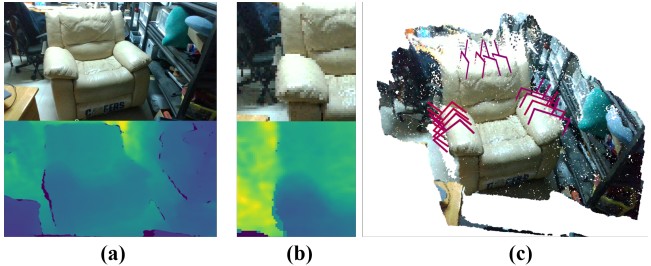

|  (a)  |  (b)  |  (c)  |

Figure 10: **Grasp detection in a room-scale scene.** (a) Target RGB image and depth image of an ordinary indoor scene with a sofa placed. (b) Extracted patch RGB and normalized z image. (c) Detected grasps of the sofa.

obtain the object segmentation mask in the original dataset and adopt a Gaussian kernel to dilate the mask image for covering possible graspable regions as much as possible. Then a fixed number of patch center candidates are randomly sampled on the dilated mask image and cropped. Also, as is mentioned in [33], depth information is important to generate collision-free grasp detection results. Thus, we add domain randomization to the patch center depth on the RGBD image planes, which provides a patch-based dataset covering more of the possible circumstances and improves the robustness of our training method.

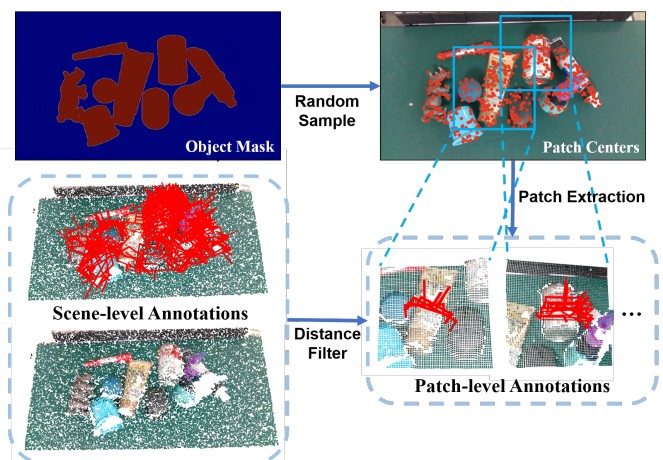

Figure 11: **Pipeline of Normalized Patch Dataset Preparation.**

**Grasping Order and Execution in Clutters:** In our cluttered scene clearance experiments, we mainly adopt the perception-planning-grasping open-loop scheme. For each grasp attempt, firstly, the robot moves to the initial home pose and captures the single-view RGBD image, from which multiple normalized grasp regions are extracted. Then, grasps are detected in each region to compose the scene-level grasps for later grasp execution. During the grasping part, the grasp with the highest score is selected as the current target pose. We utilize the MoveIt [34] framework to plan the robot's trajectory and execute the grasp poses using the UR5e equipped with the Robotiq gripper. RRT [35] serves as our path planner to plan a feasible trajectory to the target pose and execute the final grasp action. To avoid potential collisions and unsafe planning trajectories, MoveIt's collision box construction tool is employed to define collision geometry. In case of failed grasp execution resulting from collisions or inaccurate grasps, the scene will be restored, and the objects will be re-placed.

**Target-oriented Grasping:** Actually, by utilizing the proposed region-aware grasp framework, we successfully decouple the grasp location problem and grasp detection problem. As we state in the closed-loop grasping section in the manuscript, with RNGNet, we are able to detect grasps in any region regardless of the patch location or the scene structure. This implies a practical downstream

application potential when there is no need to generate grasps across the scene. In short, Region-aware Grasp Framework allows a more flexible patch localization for different purposes.

**Closed-loop Grasping Pipeline:** As demostrated in the Algorithm. 1, we integrate grasp detection with simple closed-loop robot controlling and keep tracking the real-time grasp detection results to achieve closed-loop grasping in dynamic scenes.

---

**Algorithm 1** Closed-loop Grasping in Dynamic Scenes

---

**Inputs:**
$M_{workspace}$ - workspace mask
$min\_dis$ - distance threshold
**Python-Style Pseudocode:**

```
1:  ## Tracking phrase
2:  while dist(grasps[0].pose, robot.pose) > min_dis do
3:      image = get_RGBD_image()
4:      centers = uniform_sample(M_workspace)
        # sample patch centers within the mask
5:      patches = normalize(patch_extract(centers, image))
        # extract and normalize patch regions
6:      grasps = RNGNet(patches)
        # detect grasps for objects in the workspace
7:      grasps.sort_by_score()
8:      robot.set_speed(grasps[0].pose − robot.pose)
        # robot moving towards the best grasp pose
9:  end while
10: ## Grasping phrase
11: robot.move_pose(grasps[0].pose)
    robot.grasp()
    # move robot to the final grasp pose and grasp
```

---

**Hyperparameters and Loss:** In the patch extraction, we extract normalized patches with size $S \times S = 64 \times 64$ by default. We process the input image as the same size $640 \times 360$ and adopt the pre-trained Grasp Heatmap Model checkpoint in [15] for a fair performance comparison. In the feature extraction part, we directly adopt the ResNet-18 [36] with half width as the mainstream of our backbone in RNGNet. For the implementation of the Point-wise Gate module, a 3-layer shared MLP with the same width as the corresponding feature maps is adopted to generate gate value for feature maps of each resolution. In our rotation heatmap predictor, the settings about rotation anchor are kept unchanged from those in the [15]. AdamW optimizer is adopted to train our RNGNet.

The training loss of RNGNet is the weighted sum of the regression losses and the anchor classification losses:

$$L = L_{\theta\_cls} + L_{\theta\_reg} + L_{\gamma\beta} + L_t + L_w, \tag{7}$$

where $L_{\theta\_cls}$ and $L_{\gamma\beta}$ are the losses for Euler angle classification, calculated using focal loss [37] and $L_{\theta\_reg}$, $L_t$ and $L_w$ are the losses for $\theta$ refinement, translation regression and width regression, calculated using smoothed L1 loss.

## C  Discussion

**Data Efficiency:** To further confirm the data efficiency of our framework, we conduct extra experiments by training methods with different portions of data. For fair comparisons, we test our method without Grasp Heatmap Model (pretrained Grasp Heatmap Model from HGGD uses all the training data and may lead to higher test results), that is, using the farthest sampling in the filtered foreground pointcloud to extract local regions. As is shown in the Table 12 below, our method achieves better data efficiency than other baselines. Even with only 10% training data, our method performs a competitive grasp detection performance.

| Method | 1/50$^\dagger$ scenes | 1/20$^\dagger$ scenes | 1/10$^\dagger$ scenes | Full scenes |
|---|---|---|---|---|
| GSNet | 8.64 | 30.65 | 36.15 | 47.81 |
| AnyGrasp | 7$^\ddagger$ | - | 35$^\ddagger$ | 49.01 |
| RNGNet *w/o* Heatmap | **25.60** | **41.61** | **47.43** | **58.13** |

$^\dagger$ "1/X scenes" means annotations from X scenes are used out of total 100 training scenes.
$^\ddagger$ Approximate results from Fig. 15 (Evaluation results on the GraspNet-1Billion test set when training with different portions of real-world data) in the AnyGrasp paper[9].

Table 12: **Experiments of Method Data Efficiency.** Showing APs on all Realsense test split with different portions of training data.

**Equivariant Neural Networks:** The proposed RNGNet is not inherently equivariant. We have incorporated rotation augmentation techniques to reduce the burden on the network to learn equivariance from scratch, thereby improving generalization and reducing the amount of training data required. Experiments above also show the efficacy of our proposed Region-aware Grasp Framework for offering better sample efficiency than former methods. In the future, to further improve the performance and efficiency of our approach, we may use equivariant networks in our framework.

**Introduction of Scene Priors:** Actually, our approach is designed to be easily integrated with widely adopted preprocessing modules to enhance results and extend functionality. Such shape completion and instance segmentations can serve as the data processing module for the input of our Region Normalized Grasp Network. Shape completion enhances object surface information, contributing to collision-free and stable grasp detection, while instance segmentation can provide valuable object pose information for more precise grasp detection. In practice, we have also explored integrating SAM (Segment Anything Model) [38] as the instance segmentation model and Grounding Dino [39] as the object detection model, enabling our method to address target-oriented and instruction-guided grasping tasks seamlessly.

# D   Limitations

Although our proposed region-aware framework demonstrates excellent performance and flexibility across diverse grasping tasks, several limitations remain. Primarily, our method relies on single-view RGBD images, which are vulnerable to degradation from severe noise and occlusion. Secondly, our current design focuses exclusively on common rigid bodies, and does not address the challenges posed by transparent, reflective, or deformable objects. Lastly, while our closed-loop algorithm addresses grasping tasks in some dynamic scenes, our online grasp generation and selection methods could be further optimized. For example, integrating it with reinforcement learning could enhance adaptability, facilitating smoother task execution.

