# OpenReview forum: "Region-aware Grasp Framework with Normalized Grasp Space for Efficient 6-DoF Grasping"
_robot-learning.org/CoRL/2024/Conference — CoRL 2024_

### Official Review · Reviewer_yaiB · 2024-06-28
**Interesting idea, yet no real world experiments to support it**

**Originality:** 3
**Technical Quality:** 2
**Clarity Of Presentation:** 4
**Potential Impact:** 3
**Recommendation:** 4
**Confidence:** 4

**Review:**

[strengths]
(relevance/significance) The paper addresses an important question: quickly generating 6 DoF grasps in cluttered scenes.
(originality) The main novelty is in normalizing the input image to be scale, translation, and rotation invariant.
(quality) In the real world, the authors successfully clear cluttered scenes using the proposed approach, and showcase ability to pick up a moving object by running closed loop grasping.
(clarity) The paper is written in a clearly and easy to understand

[weaknesses]
(real world experiments) Authors don't ablate their method on the real world task, and show marginal improvement over baseline (Table 5)

The authors propose two novelties: normalizing space and a pointwise gate (PG) module as a part of neural network.
Through ablation of the normalizing space, authors show significant yields in average precision (AP), while PG has a marginal improvement in AP.

The authors provide no real world experiments/ablations for either of the proposed novelties to show that this translates to improvement on the real robot, and if so, by how much! This is my main problem with this paper. The authors provide no evidence that their novelties translate to improvement in the task of clearing the table. If the authors are able to show this, I believe it would be a much stronger paper. Further, in the real world experiments, the authors show only a marginal improvement over AnyGrasp baseline. Both methods are able to clear 7 tested scenes, but for RNGNet (authors) it took 60 grasp attempts, and 65 for AnyGrasp.

Picking up on conveyor belt was a nice demo that shows fast inference of their approach (the video had a significant multiplier though).
The authors should state limitations of the approach in the main paper, not leave them for appendix.

EDIT: The authors later provided real world experiments and ablations showing their approach is superior to the baseline. Furthemore, all of my below questions are answered and I vote this paper gets accepted.

**Quality Of The Limitations Section:**

1

**Questions For Rebuttal:**

Can you provide real-world experiments/ablations for either of the proposed novelties to show that the novelties translate to improvement on the real robot, and if so, by how much? No AP, but PPM (picks per minute) or number of attempts to clear the table.

Further, in real-world experiments, can you show significant improvement over the AnyGrasp baseline?

How do you ensure the reachability of your grasps? How is this accounted for in the experiments?

What are the real limitations of your method? When does your method mostly fail?

EDIT: all questions are answered, thank you

**Robotics Focus:**

4

**Summary Of Paper:**

Authors address the problem of generating grasps for a gripper in a cluttered scene. The grasps are represented as a pose and a gripper width. The main novelty is in the way the RGBD input is processed. The authors "normalize" the patches in the input to be scale, translation, and rotation invariant. Ablations on a chosen dataset show that this normalization yields significant improvement in average precision (AP) over other methods. The authors validate the method on the real robot and show they can clear the cluttered scenes, as well as pick up a moving object due to fast inference time of the proposed method.

**Summary Of Recommendation:**

My recommendation is based on real world experiments showing strength of the proposed approach: compared to SoTA as well as ablations

---

### Official Review · Reviewer_aAbj · 2024-07-21
**RNGNet: State-of-the-Art Performance in Grasp Pose Detection**

**Originality:** 4
**Technical Quality:** 5
**Clarity Of Presentation:** 4
**Potential Impact:** 4
**Recommendation:** 4
**Confidence:** 4

**Review:**

This paper did many things well, including:
- Introducing the Proposing a new Region-aware Grasp Framework, the Normalized Grasp Space representation, and the Region Normalized Grasp Network.
- Presenting a summary of relevant grasp pose detection systems, as well as recent region normalization approaches for point cloud-based grasp pose detectors.
- Illustrating the normalized grasp region patches, network architecture, and framework well with illustrations.
- Explaining characteristics of the NGS that enable the RNGNet to achieve state-of-the-art performance.
- Proposing a network architecture that fits into the proposed framework to deal with the identified issues, enabling improvements in both accuracy and efficiency.
- Presenting a detailed breakdown of performance of eight baseline systems on the GraspNet dataset for a fair comparison against existing state of the art.
- Providing an additional breakdown of performance at different scales, with a comparison against the relevant state of the art.
- Providing additional sanity check qualitative visualizations to ensure all top grasps proposed by the system appear more reasonable than the baselines.
- Performing ablation studies to ensure each operation of the NGS and each input to the RNGNet aided performance, as well as on the network architecture.
- Verifying that RNGNet outperformed two of the baseline systems when evaluated on a real robot.
- Further demonstrating the system's performance on dynamic handover and conveyor real robot tasks.
- Demonstrating in simulation how the system generalizes with grippers of different sizes.
- Was well written.

This paper could be improved by:
- Citing Rethinking 6-Dof Grasp Detection: A Flexible Framework for High-Quality Grasping by Tang et al. This recent work from 2024 outperforms the other cited works on the GraspNet dataset, but underperforms RNGNet.

**Quality Of The Limitations Section:**

3

**Questions For Rebuttal:**

The only suggestion I have for improving this paper is to cite Rethinking 6-Dof Grasp Detection: A Flexible Framework for High-Quality Grasping by Tang et al., that outperforms the other baselines but underperforms RNGNet.

**Robotics Focus:**

4

**Summary Of Paper:**

This paper identifies an inherent shortcoming of existing point cloud-based grasp pose detection neural networks, namely that the area of interest can vary in the presence of clutter. It proposes a Normalized Grasp Space to represent regions of interest and a Regional Normalized Grasp Network that operates on this space. RNGNet is trained on the GraspNet-1Billion dataset, and achieves state-of-the-art performance, both in precision and runtime on the evaluation set. It also outperforms baseline systems when evaluated on a real robot, and achieves high performance in more difficult dynamic tasks.

**Summary Of Recommendation:**

RNGNet's network design is based on an analysis of the drawbacks of existing systems. It achieves a new state-of-the-art on the GraspNet dataset in grasp detection precision while outperforming other baselines at detection speed. It is also shown to be more effective than several baseline systems in real robot experiments. The exhaustive ablation studies support the design decisions.

---

### Official Review · Reviewer_GvsS · 2024-08-01
**Using local regions for grasping. Performance improvement over existing methods.**

**Originality:** 2
**Technical Quality:** 3
**Clarity Of Presentation:** 3
**Potential Impact:** 2
**Recommendation:** 3
**Confidence:** 3

**Review:**

Strengths:
1. Simulation and real-world experiments suggest that this region-based method is effective and has qualitative improvement over existing methods.
2. Because the scales of the grasps are normalized based on the region sizes, the learned model can generalize to different object sizes. This generalization is illustrated in Figure 9 and in experiments.
3. Overall, the paper is easy to read and presents details for the implementations.

Weakness:
1. As the authors discuss in the related work section, the idea of utilizing regional 3D geometry has been applied in many grasping works before. The authors argue that most existing methods extract local regions of predetermined sizes. However, [15] extracts local regions based on predicted grasp widths, which are not fixed. As a result, the technical novelty is limited.
2. This method generates the initial heatmap for sampling candidate regions using the Grasp Heatmap Model from [15]. The Grasp Heatmap Model takes in the whole scene as input and is, therefore, not invariant to different grasp sizes, rotations, and translations.
3. Different from existing works such as EdgeGrasp, the neural network itself is not equivariant. Therefore, the neural network model still needs to learn to generate different grasps when the region feature is rotated. This reduces the sample efficiency and generalization of the method to different object orientations.

**Quality Of The Limitations Section:**

2

**Questions For Rebuttal:**

Besides the concerns above I hope the authors can address, I have the following questions:
1. I hope the authors can clarify why the method has equivariance for theta because the patch seems to be a square which is not symmetric with respect to the z-axis.
2. In Table 5, why are the different methods tested on different numbers of trials?
3. Why are grasping methods that leverage shape completion, equivariant neural networks, and instance segmentations not discussed or compared against? They are also important for grasping from cluttered table-top scenes.

**Robotics Focus:**

4

**Summary Of Paper:**

This paper proposes to normalize SE(3) grasps in table-top scenes based on local regions and uses a learning-based method to generate normalized grasps based on local features. Specifically, a heatmap indicating potential grasps is generated using a prior method [15], and the candidate regions are sampled from the heatmap and further normalized based on the gripper size, distance to the camera, and focal length of the camera. The RGBXYZ image of each candidate region is fed to a neural network to predict grasp rotations, widths, and translational offsets.

**Summary Of Recommendation:**

This paper shows good quantitative improvements over existing methods. However, leveraging local geometric regions has been widely adopted. The technical novelty needs to be further discussed.

---

### Decision · Program_Chairs · 2024-09-04

**Decision:**

Accept

**Comment:**

The reviewers are in agreement that the paper addresses an important and relevant problem. They also agree that the experiments in simulation are thorough and the experiments show some improvement over the existing literature.

However, the specific novelty of the paper appears minimal to two of the reviewers. This coupled with the only minor increase in performance on real world tasks, makes it difficult to draw a strong conclusion from the paper. There's also some question as to how the equivariance stated as a contribution is ensured.

---Update Post Rebuttal---
Thanks to the authors and reviewers for a thorough discussion of the concerns above and in the initial reviews.

Based on the updated information the reviewers and I agree that we can now recommend the paper for acceptance.

As noted by one reviewer there is remaining concern about the lack of comparison with "grasping methods that leverage shape completion, equivalent neural networks, and instance segmentation."

We also strongly suggest the authors incorporate their answers form the rebuttal phase into the paper to improve the clarity of discussions on the contributions and the technical approach.